# Saponin Fraction CIL1 from *Lysimachia ciliata* L. Enhances the Effect of a Targeted Toxin on Cancer Cells

**DOI:** 10.3390/pharmaceutics15051350

**Published:** 2023-04-28

**Authors:** Paulina Koczurkiewicz-Adamczyk, Karolina Grabowska, Elżbieta Karnas, Kamil Piska, Dawid Wnuk, Katarzyna Klaś, Agnieszka Galanty, Katarzyna Wójcik-Pszczoła, Marta Michalik, Elżbieta Pękala, Hendrik Fuchs, Irma Podolak

**Affiliations:** 1Department of Pharmaceutical Biochemistry, Faculty of Pharmacy, Jagiellonian University Medical College, 30-688 Kraków, Poland; 2Institute of Diagnostic Laboratory Medicine, Clinical Chemistry and Pathobiochemistry, Charité—Universitätsmedizin Berlin, Corporate Member of Freie Universität Berlin and Humboldt-Universität zu Berlin, 13353 Berlin, Germany; 3Department of Pharmacognosy, Faculty of Pharmacy, Jagiellonian University Medical College, 30-688 Kraków, Poland; karolina.grabowska@uj.edu.pl (K.G.);; 4Department of Cell Biology, Faculty of Biochemistry, Biophysics and Biotechnology, Jagiellonian University, 30-387 Kraków, Poland

**Keywords:** targeted toxin, targeted therapy, CIL1 saponin fraction, anti-cancer activity, endosomal escape, ribosome-inactivating protein

## Abstract

Saponins are plant metabolites that possess multidirectional biological activities, among these is antitumor potential. The mechanisms of anticancer activity of saponins are very complex and depend on various factors, including the chemical structure of saponins and the type of cell they target. The ability of saponins to enhance the efficacy of various chemotherapeutics has opened new perspectives for using them in combined anticancer chemotherapy. Co-administration of saponins with targeted toxins makes it possible to reduce the dose of the toxin and thus limit the side effects of overall therapy by mediating endosomal escape. Our study indicates that the saponin fraction CIL1 of *Lysimachia ciliata* L. can improve the efficacy of the EGFR-targeted toxin dianthin (DE). We investigated the effect of cotreatment with CIL1 + DE on cell viability in a 3-(4,5-dimethylthiazol-2-yl)-2,5-diphenyltetrazolium bromide (MTT) assay, on proliferation in a crystal violet assay (CV) and on pro-apoptotic activity using Annexin V/7 Actinomycin D (7-AAD) staining and luminescence detection of caspase levels. Cotreatment with CIL1 + DE enhanced the target cell-specific cytotoxicity, as well as the antiproliferative and proapoptotic properties. We found a 2200-fold increase in both the cytotoxic and antiproliferative efficacy of CIL1 + DE against HER14-targeted cells, while the effect on control NIH3T3 off-target cells was less profound (6.9- or 5.4-fold, respectively). Furthermore, we demonstrated that the CIL1 saponin fraction has a satisfactory in vitro safety profile with a lack of cytotoxic and mutagenic potential.

## 1. Introduction

Saponins are secondary metabolites common in higher plants. These compounds are characterized by an extremely broad spectrum of biological and pharmacological activities. From a therapeutic point of view, their adjuvant activity is particularly valuable, which ultimately led to the use of triterpene saponins in vaccines [1]. Although well-known hemolytic properties have been pointed out as the main restriction to their use in human vaccination, many efforts have been made to overcome these drawbacks. The most prominent example is the QS-21 saponin fraction from *Quillaja saponaria*, which has been approved for use in human vaccines as a key component of combination adjuvants (Shingrix^®^ for herpes zoster) [2]. Interestingly, scientific data provide evidence that the combined use of triterpene saponins with chemotherapeutic agents may also be a promising therapeutic approach [3]. The use of combination therapies aims not only to increase the efficacy of chemotherapeutics but also to reduce their doses, which in turn could improve the safety of therapy. Interest in triterpene saponins for targeted therapies is growing [4,5]. One such example is targeted therapy using targeted toxins. These consist of a toxic enzyme coupled to a specific cell-binding domain that targets antigens associated with cancer. The most commonly used toxins are saporin, ricin-A-chain (RTA), or dianthin [6]. Although targeting toxins in cancer treatment is a promising approach, the non-specific toxicity related to uptake by off-target cells results in numerous side effects. This is due to the relatively high doses required for target cells to get enough drug into the cytosol, which is not the case at low doses because endosomal escape is insufficient. This can be overcome by using cell-penetrating peptides or endosomal escape enhancers [7,8]. The use of saponin fractions or isolated single saponins as endosomal escape enhancers can improve the efficacy of receptor-specific chimeric toxins and reduce the dose of toxins, ultimately reducing side effects [9]. Saponins applied at concentrations that are not toxic act synergistically with the target toxins and increase cytotoxicity by over 1000-fold in cell cultures. Interestingly, the effect and synergistic potency of action are strongly related to the chemical structure of saponins. Of several structurally distinct triterpene saponins that have been tested for this effect, such as hederasaponin C, glycyrrhizic acid, helianthoside 2, β-aescin, ginsenoside Rd, and quillajasaponin, only saponinum album, which is essentially a complex triterpene saponin fraction, exhibited a strong enhancement and retained ligand–receptor specificity [10].

The saponin fraction denoted CIL1 (3-O-β-D-xylpyranosyl-(1→2)-β-D-glucopyranosyl-(1→4)-[β-D-glucopyranosyl-(1→2)-]-α-L-arabinopyranoside anagalligenin B, which was used in the current study, was isolated from *Lysimachia ciliata* L. [11]. The main component of the CIL1 fraction, desglucoanagalloside B (Appendix A), belongs to a unique group of oleanane saponins that possess a completely saturated pentacyclic skeleton with a 13β28-epoxy bridge that are characteristic of the Primulaceace family. Our previous studies have shown that desglucoanagalloside B exerts cytotoxic activity in prostate cancer cell models [12]. Interestingly, in addition to its multidirectional anticancer activity, the saponin exhibited a high selectivity of action. Its effect on normal prostate cells was much weaker, confirming its satisfactory safety profile [12]. In further studies, we demonstrated the synergistic pro-apoptotic and anti-migratory activity of the saponin fraction CIL1/2 (in which desglucoanagalloside B was the main component) in models of prostate cancer cells treated with mitoxantrone. All the data collected on the CIL1/2 saponin fraction, regarding both satisfactory antitumor activity, action selectivity, and enhancement of mitoxantrone effects in prostate cell cultures in vitro, provided the rationale for further studies on descglucoanagalloside B and saponin fractions containing this compound in the context of their use in targeted therapies [3].

In the current study, we used dianthin-30, which is one of three representatives of toxins obtained from plants of the Caryophyllaceae family. Dianthin-30 is a highly active enzyme but is devoid of a cell-binding domain and thus has low cytotoxicity. Toxin dianthin-30 can be chemically coupled or genetically fused to a ligand that discern tumor-specific receptors, making the protein an attractive alternative for targeted tumor therapies [13]. In the current study, we used a targeted toxin composed of dianthin-30 fused to the EGF ligand-binding domain. Experiments were performed on two cellular models: HER14 cells, which are NIH3T3 cells stably transfected with human EGFR, and the parental NIH3T3 cells, which were used as a control.

## 2. Materials and Methods

### 2.1. The CIL1 Saponin Fraction

The CIL1 saponin fraction was isolated from the underground parts of *Lysimachia ciliata* L. ‘Firecracker’ Primulaceae. The plant material was collected from the Garden of Medicinal Plants of Jagiellonian University, Cracow, Poland. The identification of the plant species was performed by Dr. Agnieszka Szewczyk of the Department of Pharmaceutical Botany, Faculty of Pharmacy, Medical College, Jagiellonian University. A voucher sample (no. KFg/2010036) was deposited at the Department of Pharmacognosy, Pharmaceutical Faculty, Medical College, Jagiellonian University, Cracow, Poland. For the isolation of the saponin fraction CIL1, air-dried plant material was first extracted with CHCl3 and then with MeOH containing 0.5% pyridine. After the evaporation of MeOH, the dry residue obtained was divided between n-BuOH and water. The n-BuOH extract was evaporated to dryness below 45 °C. It was separated on a silica gel column (Merck Kieselgel 60; 70–230 mesh, Merck, Rahway, NJ, USA) using CHCl_3_/MeOH/H_2_O 23:12:2 as a mobile phase. The fractions were controlled by TLC (SiO_2_, CHCl_3_/MeOH/H_2_O 23:12:2; sulfuric acid + heating 5–10 min, 105 °C) and combined accordingly. The pooled CC fractions containing saponins were then further purified by preparative TLC (commercially precoated silica gel G plates; Analtech, 500 microns; CHCl_3_/MeOH/H_2_O 23:12:2; visualization: distilled water) to obtain the CIL1 saponin fraction, the composition of which was characterized by UPLC-ESI-MS/MS (Appendix A). The main component (48.3%) was identified as desglucoanagalloside B by comparing the retention time and the mass fragmentation pattern with previously isolated saponins [11].

### 2.2. Expression and Purification of DE

The Dia-EGFpET11d plasmid was transformed into *Escherichia coli* Rosetta DE pLysS (Novabiochem, Schwalbach, Germany) [14]. Cells were incubated overnight at 30 °C in Lysogeny Broth (LB) containing 100 mg/mL ampicillin. Cells were centrifuged (5 min, 3000 g) and then the pellet was dissolved in 2 L of LB containing 50 mg/mL ampicillin. Cells were grown overnight to a final optical density of 0.8. Isopropyl β-D-1-thiogalactopyranoside (IPTG, 2 mL, 1 M) was added for 3 h at 30 °C. Cell harvests were obtained by centrifugation (10 min, 5000 g, 4 °C). Cell pellets were resuspended in 20 mL phosphate-buffered saline (PBS, pH 7.4) at 20 °C. Fusion proteins were released from the FRENCH Press at 1500 psi after thawing. The solution was centrifuged (30 min, 4 °C) and adjusted with 10 mM imidazole. The construct applied to nickelnitrilotriacetate chromatography (Ni-NTA Agarose, Qiagen, Hilden, Germany). The fractions were analyzed by SDS-PAGE (12%) after elution with imidazole (20–250 mM). The fraction containing dianthin-EGF (DE) was concentrated in Amicon centrifugal filter devices (30 kDa Millipore, Eschborn, Germany). Imidazole was removed by buffer exchange (PBS) with PD10 columns (GE Healthcare, Munich, Germany). A BCA assay (Pierce, Rockford, IL, USA) was used to determine the protein concentration.

### 2.3. Determination of Enzymatic Activity

DE rRNA N-glycosylase activities were determined as previously described [14]. The target toxin (30 pmol) was mixed with hsDNA (100 mg) in acetate buffer (pH 4, 100 mM KCl). After that, the mixture was incubated at 50 °C for 1 h. Then, the mixture was centrifuged and filtered using filtration devices (a molecular weight cutoff: 5 kDa). A Nano Drop spectrometer (ND-1000, Peqlab, Erlangen, Germany) was used to measure the filtrate’s absorbance at 260 nm. An adenine calibration curve was used to determine the amount of adenine released.

### 2.4. NIH-3T3 and HER14 Cell Culture

Swiss mouse embryo NIH-3T3 cells (cells were obtained from DSMZ, the German Collection of Microorganisms and Cell Cultures) and NIH-3T3 cells transfected with human EGFR (HER14) (cells were a kind gift from Prof. E. J. van Zoelen, Department of Cell Biology, University of Nijmegen, The Netherlands) were cultured in Dulbecco’s MEM with Glutamax™ 1 (Gibco, Paisley, UK) (with 10% FCS (Fetal Calf Serum), 100 U/mL penicillin and 100 μg/mL streptomycin). Cells were grown under standard conditions (37 °C, 5% CO_2_ and 95% humidity).

### 2.5. Experimental Procedure and Viability Assay

For viability assays, the cells were seeded in 96-well plates at a concentration of 2 × 10^3^ cells per well. Then, cells were incubated with 180 μL of medium containing 3 μg/mL CIL1 saponin. After 5 min, 20 μL solutions of DE at defined final concentrations (0.000001 to 100 nM) were added. Cells were cultured for 72 h. A colorimetric test-MTT was used to evaluate the cell viability as previously described [15]. The experiments were carried out in triplicate.

### 2.6. Proliferation Assay

Cells were seeded in a 24-well plate at a density of 1 × 10^4^ /well and incubated in the presence of compounds for 72 h. Subsequently, cells were fixed for 15 min in a formaldehyde solution in PBS (3.7%). Then, cells were washed with PBS and stained with 500 μL of 0.01% crystal violet solution for 5 min. The dye staining the cells on the plates was eluted with 500 μL methanol (25% *v*/*v*) containing citric acid (1.33% *m*/*v*) and sodium citrate (1.09% *m*/*v*). The supernatant from each well was transferred to three wells in 96-well plates, and the optical density of the extracted dye was read using a spectrophotometer SpectraMax^®^ iD3 (Molecular Devices, San Jose, CA, USA) at 540 nm (A540). The inhibition of proliferation was calculated as: inhibition of proliferation (%) = (1 − (experiment A540/control A540)) × 100.

### 2.7. Apoptosis

#### 2.7.1. Flow Cytometry Analysis

NIH3T3 and HER-14 cells were seeded at a density of 5 × 10^4^/well in 6-well culture plates. The cells were then incubated with the evaluated compounds for 48 h. Next, the cells were rinsed with PBS, tripsinized, centrifuged (300 g, 4 min) and counted in a Bürker chamber. Cells were then stained with an Annexin V/7-AAD (BD Bioscience, Ann Arbor, MI, USA) using a modified manufacturer’s protocol. Briefly, the cells were resuspended to a final concentration of 1 × 10^6^ cells/mL in a binding buffer. FITC-conjugated Annexin V and 7-AAD nuclear dye were then added and incubated for 15 min at room temperature. Stained cells were analyzed by flow cytometry using a BD LS-Fortessa cytometer and BD FACS Diva Software ver. 8.1 (Becton Dickinson, East Rutherford, NJ, USA). Three independent experiments were conducted. Each experiment analyzed 1 × 10^4^ cells.

#### 2.7.2. Luminescence Detection of Caspase 3/7

Cells were seeded at a density of 1 × 10^4^ cells/well on white opaque 96-well plates. Next, cells were incubated with the analyzed compounds for 24 h. After incubation, a Caspase-Glo^®^ 3/7 assay (according to the manufacturer’s manual) was performed. The luminescence was measured on a Spectra Max ID3 microplate reader (Molecular Devices, San Jose, CA, USA). Experiments were run in triplicate. Data were analyzed and expressed as a percentage of untreated (control) cells.

### 2.8. Safety Study

#### 2.8.1. Hepatotoxicity, Neurotoxicity and Cardiotoxicity

Hepatotoxicity, neurotoxicity and cardiotoxicity were evaluated by using human hepatocellular carcinoma cells (HepG2, HB-8065), human neuroblastoma cells (SH-SY5Y, CRL-2266™) and rat cardiomyoblasts (H9c2(2-1), CRL-1446™), respectively. Cells were purchased from the American Type Culture Collection (ATCC, Manassas, VA, USA) and cultured according to the manufacturer’s instructions. Cells were seeded at an initial density of 2 × 10^3^/well in 96-well plates for cytotoxicity assays. After overnight incubation, CIL1 was added at concentrations of 1–20 μg/mL. The cells were then incubated for 72 h. MTT reagent (MTT, Sigma Aldrich, St. Louis, MO, USA) was then added to each well at a final concentration of 0.5 mg/mL for the last 4 h of incubation. The culture medium was then removed from the wells. DMSO was used to dissolve formazan crystals that appeared. A SpectraMax^®^ iD3 (Molecular Devices, San Jose, CA, USA) was used to measure the absorbance at 570 nm. The assay was performed in triplicate.

For further evaluation of cytotoxicity, CytoToxGlo, a cell membrane integrity test (Promega), was used. Assays were performed according to the manufacturer’s protocol. Cells were seeded at a density of 2 × 10^3^ cells/well into white 96-well culture plates and cultured for 24 h. The cells were then incubated in the presence of CIL1 for an additional 72 h. Triton X-100 was used as a positive control and a solvent (0.01% DMSO) was added as a negative control. To each well, CytoToxGlo™ Reagent was added. The luminescence signal was measured using a microplate reader (Spectra Max ID3, Molecular Devices) (signal 1). A lysis solution was then added and the luminescence was read again after 15 min for total signal detection (signal 2). By subtracting signal 1 from signal 2, the percentage of live cells was calculated. The signal obtained for the control condition was set to 100% and the signal for TritonX-100 was set to 0% of living cells. The cytotoxicity was calculated as 100 minus A, where A is the viability of the cells in the analyzed sample (in relation to the control). The experiments were carried out in triplicate.

#### 2.8.2. Mutagenic Potential

In medium containing sufficient histidine to support cell growth, *Salmonella typhimurium* cells were incubated with various concentrations of CIL1 for 90 min. The cultures were then transferred to a histidine-free pH-indicator medium and aliquoted into 384-well plates. Cells that revert to amino acid prototrophy within 48 h will grow into colonies, causing the pH of the medium to drop and the color to change from purple to yellow. In this study, two strains of *Salmonella typhimurium*, TA98 and TA100, suitable for the detection of base pair substitution and frame-shift mutations were used. The test procedure was provided by Xenometrix. Briefly, the test strains were grown overnight in exposure medium and then exposed to CIL1 at final concentrations of 0.1, 0.2 and 0.5 mM in 24-well plates for 90 min at 37 °C in the absence (–S9) or presence (+S9) of 4.5% phenobarbital/β-naphthoflavone-induced rat liver S9. CIL1 was dissolved in DMSO. After preincubation, cultures were diluted in indicator medium, and the contents of each 24-well culture were distributed in 384-well plates and incubated for an additional 48 h at 37 °C. The number of wells containing bacteria was then scored by counting the yellow wells. Positive controls used in the assay were 2-nitrofluorene (2-NF) at 2 µg/mL (TA98, –S9) and 4-nitroquinoline-N-oxide (4-NQO) at 0.1 µg/mL (TA100, –S9); DMSO was used as a negative control. The experiments were carried out in triplicate. To interpret the test results, the following criteria were used: (i) an increase in the number of reverting colonies compared to the solvent control (determined by dividing the mean number of positive wells at each dose by the solvent control at baseline) and (ii) dose dependence. The solvent control at baseline was defined as the mean number of positive wells in the solvent control plus one standard deviation (SD). When an increase greater than 2 times relative to baseline was observed at more than one dose with a dose response, the sample was classified as positive, while when no response >2 times the baseline and/or no dose response was observed, the sample was classified as negative.

## 3. Results

### 3.1. CIL1 Increases DE-Mediated Inhibition of Proliferation and Cytotoxicity of HER14 and NIH3T3 Cells

To investigate the ability of the CIL1 saponin fraction to enhance the effect of the DE, we used HER14 EGFR-bearing cells and untransfected NIH3T3 parent cells, which were used as controls. First, non-toxic concentrations of CIL1 were assessed. Cells were incubated in the presence of CIL1 for 72 h. Then, the cell viability rate was determined by an MTT assay. The results showed that the analyzed cells had different sensitivities to CIL1. The NIH3T3 cell line was more vulnerable. A significant cytotoxicity was observed at a CIL1 concentration of 10 µg/mL (Figure 1A, B). Due to the significant decrease in cell viability between 5 and 10 μg/mL, it was decided to test cytotoxicity over a wider range of CIL1 concentrations (5.5–9.5, 0.5 μg/mL increments). These results are included in the Appendix A. For HER14 cells, CIL1 induced significant cytotoxicity only at the highest applied concentration (20 µg/mL). Considering the results obtained, a concentration of 3 µg/mL of CIL1 was chosen for a further detailed analysis of the synergistic effect with DE, as it was safe for both cell types.

To induce an enhanced cytotoxicity and inhibit proliferation, HER14 and NIH3T3 were co-treated with CIL1 (3 µg/mL) and DE (0.00000001–100 nM). CIL1 increased the cytotoxicity and reduced the proliferation of EGFR-expressing HER14 cells as well as control NIH3T3 cells; however, the resulting enhancement factors differed between target and off-target cells. Although the enhancement factor in HER14 cells was 2200, the corresponding enhancement factor in NIH3T3 cells was only 6.9 (Table 1). Comparable results were obtained for cells treated with DE + CIL1 in a CytoToxGlo assay, which measures the changes in the integrity of cell membrane (Appendix A). For the proliferation inhibition study, the enhancement factor in HER14 cells was again 2200, and the corresponding factor in NIH3T3 cells was 5.4. Based on the obtained results, it was possible to determine the gain in receptor specificity by dividing the enhancement factor (EF) for HER14 by the EF for NIH3T3; a ratio greater than 1 represents a gain in receptor-specific cytotoxicity. In the case of DE, the gain in receptor specificity due to CIL1 in the cytotoxicity study was 320, while for proliferation it was 399 (Figure 2 and Table 2).

### 3.2. CIL1 Saponin Exhibits a Synergistic Effect with DE and Enhances Apoptosis of HER14 and NIH3T3 Cells

To confirm the previous observations, we decided to assess the mechanism of cytotoxicity. A flow cytometric analysis of cells stained with Annexin V and 7-AAD was performed to verify the apoptosis process determined by the cocktail CIL1 (3 µg/mL) + DE (0.001 or 0.01 nM). Annexin V binds to phosphatidylserine, which is translocated to the external leaflet during the early stages of apoptosis. Annexin V is used to specify the target and identify apoptotic cells. Necrotic cells were stained with 7-ADD, which intercalates with DNA from damaged cells.

The results indicated that the combination of the CIL1 saponin fraction and DE induces apoptosis in NIH3T3 and HER14 cells. A flow cytometry analysis performed after 48 h of incubation revealed that CIL1 and DE alone did not induce apoptosis, while their cotreatment significantly increased the level of Annexin V positive cells (Figure 3A). However, the pro-apoptotic effect was definitively more pronounced in HER14 cells compared to NIH3T3 cells. The toxin used at a concentration of 0.001 nM in combination with CIL1 increased the level of apoptotic cells in NIH3T3 to 3.5%, while in HER14 it was increased to 49.3%. In addition, the level of the active form of caspase in the cells was detected luminometrically. The results show that in cells treated with toxin alone or CIL1 alone, the caspase 3/7 level did not change significantly compared to the control conditions. Coadministration of DE at a concentration of 0.01 nM with CIL1 increased the levels of executive caspases 3 and 7 in HER14 to 58%, while in NIH3T3 it was increased up to 16% (Figure 3B).

### 3.3. CIL1 Revealed a Satisfactory Safety Profile Confirmed by an In Vitro Cytotoxicity Analysis and a Mutagenicity Assay

The exclusion of hepatotoxicity, neurotoxicity and cardiotoxicity and the determination of safe concentrations of active compounds are prerequisites for the analysis of in vitro safety studies. In the present study, we investigated the initial cytotoxicity using well-established commercially available models that are widely used for preclinical safety evaluation, such as SHSY-human neuroblastoma, HepG2-human hepatoma and H9c2-rat cardiomyoblasts [16,17]. Two assays investigating the different mechanisms of cytotoxicity were used in this study. Cells were treated with the CIL1 saponin fraction in the concentration range of 1–20 µg/mL for 72 h. The results showed that SHSY-5Y was the most sensitive cell line to CIL1. At the highest concentration of CIL1, the cell viability decreased to 20%. HepG2 and H9c2 cell lines were less vulnerable to CIL1 at the highest concentrations, exhibiting a decreased viability to 51% and 39%, respectively (Figure 4A). Based on the experiments carried out, it was established and confirmed in two independent assays that CIL1 concentrations of 1–5 µg/mL are non-toxic and can be used for further analyses. An important element of safety analyses is the exclusion of the mutagenic potential of the substances tested. For this purpose, a commercially available Ames test was used in the study. The test uses specifically modified strains of Salmonella typhimurium strains (which carry mutations in genes involved in histidine synthesis). They are auxotrophic mutants; Ames tests investigate the capability of the tested substance (potent mutagen) to create mutations that result in a return to a prototrophic state so that cells can grow in a medium without histidine. We performed an experiment to explore the mutagenic effect of not only CIL1, but also its metabolites. To do so, we used the Aroclor 1254-induced S9 fraction. No mutagenic effect was observed for CIL1 (Figure 4B). The number of revertants observed in TA98 and TA100 Salmonella typhimurium cultures while being exposed to CIL1 remained low (Figure 4B). Simultaneously, the mutagenic potential of CIL1 was not altered by the metabolic activation of the bacteria (+S9) (Figure 4B). Only after exposure to reference substances (mutagenic factors-2-nitrofluorene (NF-2) at 2.0 μg/mL and 4-nitroquinoline-1-oxide (NQNO4) at 0.1 μg/mL) was a significant increase in the number of revertants observed.

## 4. Discussion

Triterpene saponins are a class of phytochemicals that are characteristic for higher plants, microorganisms and, less commonly, marine organisms [18]. The spectrum of their biological activity is broad, from adjuvant, antiviral and antimicrobial effects to cell membrane permeabilization. Saponins also exert a wide range of anticancer effects, the mechanisms of which have been the subject of detailed analyses in recent years [19,20]. These compounds have been shown to inhibit cancer cell proliferation, exhibit pro-apoptotic activity, and inhibit the invasive potential of cancer cells in vitro and in vivo. In addition, they can modulate the tumor microenvironment, which is important from the point of view of the cancer cell energy metabolism [21]. In summary, saponins influence signaling pathways and processes at different stages of cancer promotion and progression [3,22]. Despite the significant progress made in recent years, the use of saponins as anticancer agents has some flaws, predominantly due to their toxicity and low bioavailability [23,24]. An interesting option for the future clinical use of saponins is their potential in combination therapies. Many reports indicate that saponins, when used in non-toxic concentrations, can exert adjuvant activity by sensitizing the cell to chemotherapeutics or, in the case of toxins, enhancing their action [3,25]. In both cases, the key feature is to reduce the dose of the biologically active compound (chemotherapeutic agent or targeted toxin), which in turn can significantly reduce the overall side effects of therapy. Studies on the coadministration of saponins with chemotherapeutic agents are of interest to researchers, but the molecular mechanisms of these interactions are still not well understood.

Targeted toxins, consisting of tumor-selective ligands coupled to toxins, demonstrate a new class of anticancer agents with broad benefits for patients. Unfortunately, targeted therapy has some limitations, including the lack of effective cytosolic uptake, nonspecific toxicity and, finally, concomitant vascular leaks [26]. Despite these difficulties, extensive research in the field of the safety of targeted toxins has eventually led to the development of several drugs for cancer treatment that are currently available on the market. To date, the FDA has approved three targeted toxins for clinical use: Denileukin diftitox (Ontak^®^) for the treatment of T cell lymphoma [27], Moxetumomab pasudotox (Lumoxiti^®^) for the treatment of hairy cell leukemia (HCL) [28] and Tagraxofusp-erzs (Elzonris^®^) for blastic plasmacytoid dendritic cell neoplasm (BPDCN) treatment [29]. In addition, a series of other targeted toxins to treat cancers are currently being tested in clinical trials, none of these, however, have yet been approved. Several ways to address toxicity have been proposed, for example, the use of cell-penetrating peptides. One other alternative interesting solution is the use of saponins, which themselves do not have the ability to reach target cells but can enhance the uptake of targeted toxins [8]. To date, studies using both saponin fractions and isolated compounds have been successfully tested in vitro or in vivo [30,31,32,33,34].

CIL1 is a purified saponin fraction derived from *Lysimachia ciliata* L. (Primulaceae). Its components are triterpene monodesmosides and represent a rare structural group with a saturated oleanane skeleton and a 13β28-epoxy bridge (Appendix A) [11]. These saponins are found almost exclusively in the Primulaceae and Myrsinaceae families. One of the main components of the CIL1 saponin fraction is desglucoanagalloside B. Our previous studies have demonstrated the multidirectional anti-cancer activity of desglucoanagalloside B in prostate cancer cell models. We also showed that another saponin fraction containing desglucoanagalloside B (CIL1/CIL2), used together with mitoxantrone (MTX), had synergistic viability decreasing, growth inhibiting, pro-apoptotic and anti-migration effects on androgen-independent prostate cancer cells derived from prostate cancer metastases to the brain (DU-145) and bone (PC-3) [3,35]. The selectivity of action of CIL1/CIL2+MTX, confirmed in normal prostate cells (PNT2), encouraged further investigations to use CIL1/CIL2 as a potent adjuvant in targeted therapy. Furthermore, to date, there have been no reports on the use of similar structures in targeted therapies. To this end, we decided to test whether the undirected enhancer effect of the CIL1 saponin fraction, of which desglucoanagalloside B is the main component, could increase the specific cytotoxicity of receptor-targeted dianthin-30 (DE).

Dianthin-30 belongs to ribosome-inactivating proteins (RIPs) of type 1. Type 1 RIPs exert high cytotoxicity inside the cell, but only low cytotoxicity outside the cell due to the lack of a cell binding domain, making them interesting alternatives for targeted cancer therapies [13]. A study of potential therapeutic antitumor effects of one of the main components of *Saponinum album*, saponin SA1641, used in combination with targeted toxins consisting of EGF as the binding ligand and different toxins, showed that DE was one of the most effective [14]. In another study, cotreatment with SO1861, a saponin isolated from the roots of S. officinalis L., together with chemical conjugates of dianthin-30 and antibodies (cetuximab, panitumymab and trastuzumab), was investigated. Although the three immunotoxins alone did not show any cytotoxic effects in the target cells at concentrations of up to 10 nM, a strong cytotoxicity was observed in combination with SO1861 in all cases (IC_50_ values of 0.0053 nM for dianthin-30-cetuximab, 0.0015 nM for dianthin-30-panitumumab in H116 cells and 0.023 nM for dianthin-30-trastuzumab in BT-474 cells) [36]. The efficacy of dianthin was also evaluated using in vivo models. In a nude mouse model of subcutaneous HCT-116 tumors, His-tagged dianthin-30 fused to EGF was tested with the saponin SO1861 [34]. Following this treatment, a 96% reduction in tumor volume and complete regression was observed in mice [34]. Dianthin-30-EGF with saponin SO1861 was also used to treat pancreatic BxPC-3 cell carcinoma in a nude mouse model. The results showed that the cotreatment with dianthin-3-EGF and saponin SO1861 reduced the tumor volume by 97%, with most mice showing complete regression [32].

Our results showed that non-toxic concentrations of the CIL1 saponin fraction can enhance DE cytotoxicity against tumor cells expressing large amounts of target receptor (HER14 cell line) while off-target cells—untransfected NIH3T3—are not affected. The data presented, including cytotoxicity analysis showed that the safe (and at the same time active) concentration of saponins is below 3 µg/mL. These data coincide with other data from the literature, where the authors have confirmed the safety of other saponins at similar levels [37]. Incubation of HER14 cells in the presence of DE alone resulted in an IC_50_ value of 0.54 nM, while the addition of CIL1 decreased the IC_50_ value to 0.000245 nM. In target-receptor-free NIH3T3 cells, incubation in the presence of DE alone resulted in an IC_50_ of 55.2 nM and with CIL1 the value was 8.1 nM. Enhancement factors allowed estimation of the EGFR-dependent influence on the enhanced cytotoxicity mediated by the CIL1 fraction. The presence of EGF as a ligand may also play an important role in the amplification of the effects of CIL1 saponins. The observation that the enhancement factor for targeted cytotoxicity is greater for HER14 (2200) than for NIH3T3 (6.9) may indicate the specificity of the ligand receptor. The gain in receptor specificity broadens the therapeutic window and thus facilitates the development of treatment regimens. However, to further analyze the ability of the CIL1 saponin fraction to enhance ligand-specific cytotoxicity, the same experiments should be performed with nontargeted dianthin without the EGF ligand [10]. In the current study, we focus on testing whether the enhancement of the toxin effect by CIL1 will be transferred to other physiologically important cellular processes. We first checked whether DE + CIL1 have an influence on cell proliferation. The results of the experiment show that CIL1 significantly increases proliferation-inhibiting activity. Although the effectiveness coefficient for the proliferation assay is similar to that obtained in the cytotoxicity assay (0.34 and 0.54 nM, respectively), it is noteworthy that the CIL1 fraction amplifies the effect of the toxin, so that a lower dose of the toxin is needed to effectively inhibit proliferation compared to the dose that has a cytotoxic effect. Additionally, in the case of apoptosis induction, this effective dose is even lower, as a pro-apoptotic effect is observed already at a toxin concentration of 0.001 nM. This effect is very interesting and valuable. Cells incubated with saponin alone or DE alone do not show an enhanced apoptotic effect. The DE + CIL1 cocktail induces apoptosis in 50% of the cell population. Subsequent experiments showed a stronger effect of the DE + CIL1 combination against HER14 cells compared to NIH3T3 cells.

The safety of the CIL1 saponin fraction has been the subject of a detailed analysis in the present manuscript. Many authors have pointed out the possibility of saponin-induced toxic effects [38]. The selection of a non-toxic dose was an important element in confirming safety. Furthermore, the cytotoxicity of the cell lines included in the models of cardiotoxicity, hepatotoxicity and neurotoxicity was investigated. Each model confirmed the safety of CIL1 at the concentration used in the current study. Moreover, we excluded the possible mutagenic potential of CIL1, thus confirming its safety. In fact, most data available in literature exclude the mutagenic effect of saponins [39]. As our studies have a preliminary character, we did not at this stage establish the hemolytic potency of fraction CIL1 on human erythrocytes. Generally, triterpene saponins have lesser hemolytic effects than steroid saponins. Literature data on some compounds of similar structure with a 13β28-epoxy-oleanane skeleton indicate a HD_100_ value of 12–55 μM [40]. Nevertheless, as pointed out by many authors, extrapolation of results based only on selected structural features may be misleading, as not only do the functional groups and the glycidic moieties themselves affect the hemolytic activity of saponins, but so does their overall conformation. The mechanism underlying hemolysis by saponins is still unclear; the most common hypothesis is that saponins interact with cholesterol in the membranes of erythrocytes, forming pores that destabilize the membrane.

The results obtained in the current study highlight, for the first time, the potential of saponins present in the CIL1 fraction for dianthin targeted therapy. There appears to be the potential for an even greater reduction in toxin doses, which may translate into the greater safety of anticancer therapies.

## Figures and Tables

**Figure 1 pharmaceutics-15-01350-f001:**
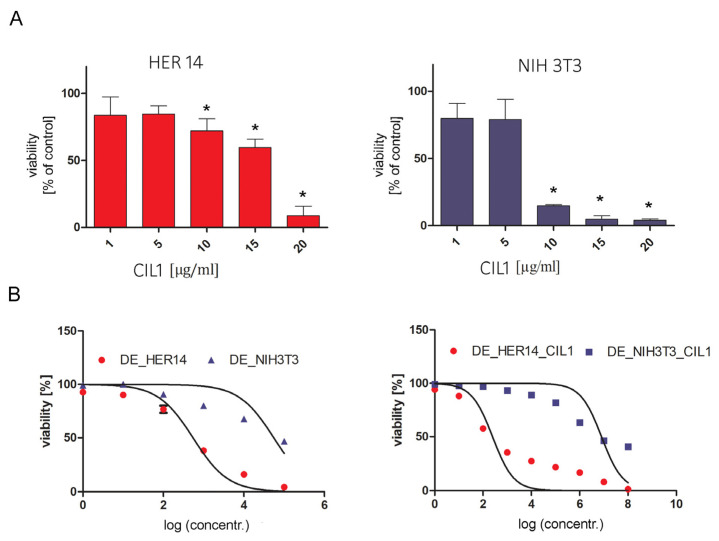
Influence of CIL 1 on the combined induction cytotoxicity by DE in HER14 and NIH3T3 cells. The non-toxic concentration of CIL1 (3 µg/mL) confirmed in the MTT assay (**A**) was used in the study. HER14 or NIH 3T3 cells were seeded at an initial density of 2 × 10^3^ cells/well in 96-well plates. Cells were incubated for 72 h with DE in the presence or absence of CIL1. Cell viability was determined using an MTT assay (**B**). Each experiment was repeated in triplicate. Statistical significance (*) was calculated relative to control (*p* < 0.05) using Brown Forsythe and Welsch ANOVA test, along with a post-hoc unpaired t-test with Welch’s correction.

**Figure 2 pharmaceutics-15-01350-f002:**
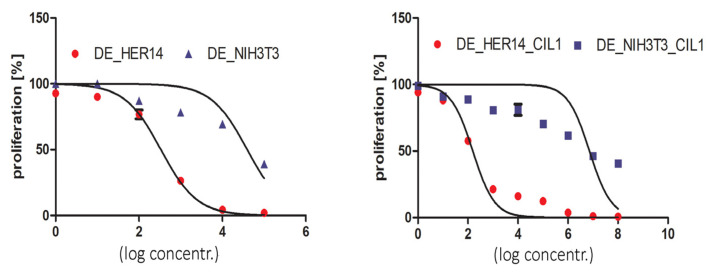
Influence of CIL 1 on the combined induction of DE proliferation inhibition in HER14 and NIH3T3 cells. HER14 or NIH 3T3 cells were seeded at an initial density of 2 × 10^3^ cells/well in 96-well plates. Cells were incubated for 72 h with DE in the presence or absence of CIL1 (3 µg/mL). Cell proliferation was measured using a crystal violet assay. Each experiment was repeated in triplicate.

**Figure 3 pharmaceutics-15-01350-f003:**
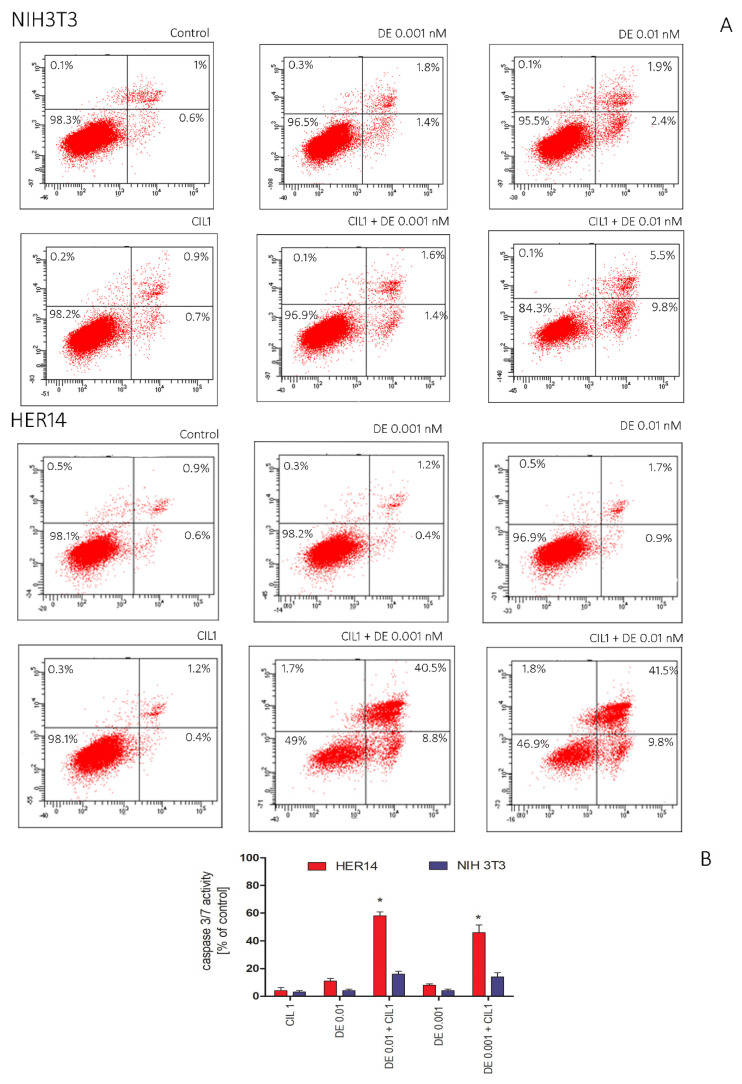
Influence of CIL 1 on the combined induction of apoptosis by DE in HER14 and NIH3T3 cells. Cells were incubated for 48 h with DE in the presence or absence of CIL1 (3 µg/mL) (flow cytometry analysis) or 24 h for luminescence analysis. Cell apoptosis was detected using (**A**) Annexin V/7-AAD staining followed by flow cytometry analysis and (**B**) luminescent detection of caspase 3/7. Each experiment was repeated in triplicate. Statistical significance (*) was calculated relative to control (*p* < 0.05) using Brown Forsythe and Welsch ANOVA test, along with a post-hoc unpaired *t*-test with Welch’s correction.

**Figure 4 pharmaceutics-15-01350-f004:**
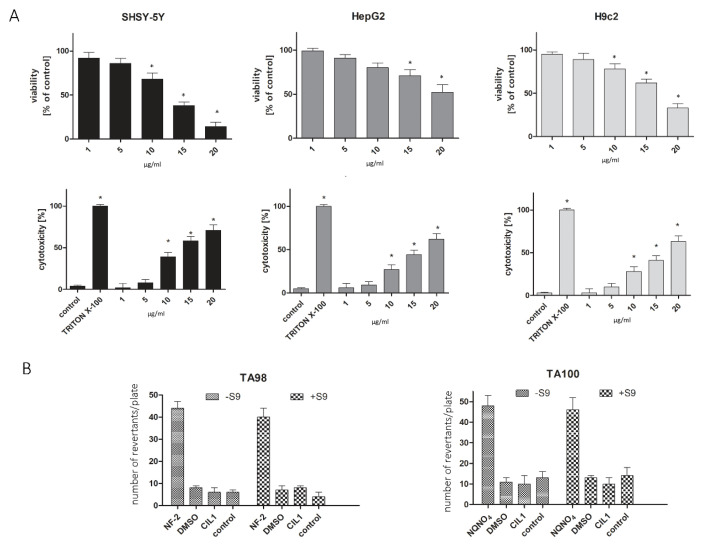
Cytotoxicity (**A**) and mutagenicity (**B**) of CIL1. The neurotoxicity, hepatotoxicity and cardiotoxicity of CIL1 were evaluated using the following cell lines: human neuroblastoma (SHSY-5Y), human hepatocellular carcinoma (HepG2) and rat cardiomyoblast (H9c2). Cells were incubated in the presence of CIL1 in the concentration range of 1–20 µg/mL for 72 h, then MTT and CytoTox Glo assays were performed. The graphs represent the mean percent of viability/cytotoxicity ± SD compared to untreated cells (control) (**A**). Each experiment was repeated in triplicate. The mutagenicity was evaluated using the Ames test. *Salmonella typhimurium* strains TA98 and TA100 were used in the study in the absence (−S9) or presence (+S9) of the S9 fraction. Bacterial cultures were incubated for 48 h with CIL1 (10 µg/mL) or positive controls 2-nitrofluorene (NF-2) at 2.0 μg/mL and 4-nitroquinoline-1-oxide (NQNO_4_) at 0.1 μg/mL (**B**). Each bar represents the average number of revertants/plate (± SD). Statistical significance (*) was calculated relative to control (*p* < 0.05) using Brown Forsythe and Welsch ANOVA test, along with a post-hoc unpaired *t*-test with Welch’s correction.

**Table 1 pharmaceutics-15-01350-t001:** Influence of CIL 1 on the combined induction cytotoxicity by DE in HER14 and NIH3T3 cells.

	IC_50_ (nM)	EF
DE_HER14	0.54	-
DE_NIH3T3	55.2	-
DE_HER14_CIL1	0.000245	2200
DE_NIH3T3_CIL1	8.01	6.9

The table shows the IC_50_ values (nM) and the saponin-mediated enhancement factors (EF) for DE in HER14 and NIH3T3 cells. The receptor specificity brought about by CIL1 was therefore increased 320-fold.

**Table 2 pharmaceutics-15-01350-t002:** Influence of CIL 1 on the combined induction of DE proliferation inhibition in HER14 and NIH3T3 cells.

	IC_50_ (nM)	EF
DE_HER14	0.34	-
DE_NIH3T3	39.4	-
DE_HER14_CIL1	0.000156	2179
DE_NIH3T3_CIL1	7.21	5.4

The table shows the IC_50_ values (nM) and the saponin-mediated enhancement factors (EF) for DE in HER14 and NIH3T3 cells. Thus, the receptor specificity caused by CIL1 was increased 399-fold.

## Data Availability

Data will be made available on request.

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
