# Peer review of "Saponin Fraction CIL1 from Lysimachia ciliata L. Enhances the Effect of a Targeted Toxin on Cancer Cells"

_pharmaceutics, 2023, doi:10.3390/pharmaceutics15051350_

Round 1

Reviewer 1 Report

In this manuscript, the authors investigated the effect of cotreatment with the saponin fraction CIL1 of Lysimachia ciliata L + EGFR-targeted toxin dianthin (DE) on cell viability (using MTT) assay, proliferation, and pro-apoptotic activity. They claimed to found that cotreatment enhanced target cell-specific cytotoxicity and antiproliferative and proapoptotic properties. Moreover, the authors also reported that the CIL1 saponin fraction exhibited satisfactory in vitro safety. Overall, this is a well-written manuscript, however, the authors need to conduct further studies to support their results. Especially, for cytotoxicity and apoptosis studies, authors must provide the fluorescence/confocal images of the cells using (appropriate stains/dyes) to support the claims. 

Author Response

Thank you very much for your comments on our manuscript. We appreciate your careful and kind assessment. We already planned the experiments in such a way that the final results would be confirmed by two independent methods that investigated different mechanisms (cytotoxicity – metabolic activity evaluation and additionally: cell membrane integration; apoptosis – annexin V expression and level of caspase in the cells). When planning experiments, we focused in this manuscript on quantitative methods, resulting more from the analysis of a representative population of cells than single cell analysis.  To confirmed the effect of cytotoxicity table which contain results from CytoToxGlo assay (cell membrane integration mechanism) were included in Supplementary Materials (Table 1S). We agree with the Reviewer that additional photos would enrich the work and in the further stages of the development of the subject we will pay attention to it. Nevertheless, in the present manuscript, the effect of DE + CIL1 was proven by two biological parameters with two independent methods each, which is to our opinion completely sufficient as evidence.

Reviewer 2 Report

The authors of this study made efforts to investigate the potential of saponins in combination therapies for cancer treatment. They provided an overview of the biological activity of saponins and their anti-cancer effects, as well as their drawbacks, including toxicity and poor bioavailability. The authors also discussed the potential use of saponins in combination therapies, which can reduce the overall side effects of therapy. They further highlighted the limitations of targeted toxins in cancer treatment and proposed the use of saponins as an interesting solution to enhance the uptake of targeted toxins. The authors conducted studies on a purified saponin fraction obtained from Lysimachia ciliata L. and demonstrated its specificity, selectivity, and versatility of action on prostate cancer cells, which showed synergistic cytotoxic, cytostatic, pro-apoptotic and anti-invasive effects when used in combination with mitoxantrone. Overall, the authors made significant efforts to investigate the potential of saponins in combination therapies and provided evidence to support their claims.

Author Response

Thank you very much for your comments on our manuscript. We appreciate your careful and kind assessment.

Reviewer 3 Report

This study is not in my field of interest. However, I do have some comments. The study is characterized by a careful study using UPLC of the composition of the saponin extract used (CIL1). Since this identification of the substituents is only based on MS the compounds are not unequivocally identified. It would have been interesting to get an estimation of the concentration of the major components.

I, however, have a problem with the number of repetitions and if the experiments have been performed in duplicate or triplicates.

No standard deviation is given in Figure 2? I do not understand this figure. I assume the decreased IC50 is caused by the addition of CIL1. I do not, however, see to what concentration CIL1 has been added.  Neither do I see how many times the experiments have been repeated and if the experiments were performed in triplicate. I believe it is of utmost importance that the authors have ensured reproducibility.

I do not see how many times the cellular assay has been performed. In Figures 3 and 4. Average values have been calculated and standard deviation included indicating repetitions of the experiments.

In the luminescence study is reported that the experiment was performed twice in duplicate. I would have preferred three times in triplicate.

Saponin's main characteristic is that they hemolyze living mainly because they reduce surface tension. They can solubilize fats. This effect generally is nonspecific and may reduce the possibility of using saponin as a drug.  It is surprising that the authors have not mentioned this definition of saponins and have not called attention to the fact that the biological effect could be caused by damage to the cell membranes.

It would be interesting to see if any single compound purified from the fraction CIL1 could mimic some of the effects described in this manuscript. 

This study is not in my field of interest. However, I do have some comments. The study is characterized by a careful study using UPLC of the composition of the saponin extract used (CIL1). Since this identification of the substituents is only based on MS the compounds are not unequivocally identified. It would have been interesting to get an estimation of the concentration of the major components.

I, however, have a problem with the number of repetitions and if the experiments have been performed in duplicate or triplicates.

No standard deviation is given in Figure 2? I do not understand this figure. I assume the decreased IC50 is caused by the addition of CIL1. I do not, however, see to what concentration CIL1 has been added.  Neither do I see how many times the experiments have been repeated and if the experiments were performed in triplicate. I believe it is of utmost importance that the authors have ensured reproducibility.

I do not see how many times the cellular assay has been performed. In Figures 3 and 4. Average values have been calculated and standard deviation included indicating repetitions of the experiments.

In the luminescence study is reported that the experiment was performed twice in duplicate. I would have preferred three times in triplicate.

Saponin's main characteristic is that they hemolyze living mainly because they reduce surface tension. They can solubilize fats. This effect generally is nonspecific and may reduce the possibility of using saponin as a drug.  It is surprising that the authors have not mentioned this definition of saponins and have not called attention to the fact that the biological effect could be caused by damage to the cell membranes.

It would be interesting to see if any single compound purified from the fraction CIL1 could mimic some of the effects described in this manuscript. 

Author Response

This study is not in my field of interest. However, I do have some comments. The study is characterized by a careful study using UPLC of the composition of the saponin extract used (CIL1). Since this identification of the substituents is only based on MS the compounds are not unequivocally identified. It would have been interesting to get an estimation of the concentration of the major components.

I, however, have a problem with the number of repetitions and if the experiments have been performed in duplicate or triplicates.

Thank you for your valuable remark. All results included in the manuscript come from three independent experiments performed in triplicates. Due to the fact that it raised doubts - both in the materials and methods section and in the caption under the figures, such information was included.

No standard deviation is given in Figure 2? I do not understand this figure. I assume the decreased IC50 is caused by the addition of CIL1. I do not, however, see to what concentration CIL1 has been added.  Neither do I see how many times the experiments have been repeated and if the experiments were performed in triplicate. I believe it is of utmost importance that the authors have ensured reproducibility.

Thank you for your valuable comments. Graph on the left side show the effect of DE used alone on cellular proliferation of two cell lines (NIH3T3-control cells, without expression with EGF and HER14 cells – overexpressed EGF), graph on right side show the effect of cotreatment DE+CIL1 on cellular proliferation of both cell lines. The figure aims to present the enhancement of the toxin effect in the presence of CIL1, and this effect is selective (stronger against EGF-possessing cells). The lack of standard deviation on the graph is due only to aesthetic reasons, each experiment was carried out three times, in each experiment three independent samples were analyzed. In order to improve the informativeness of the figure, the saponin concentration was added to the description under the figure, and the description was further detailed with the missing information.

I do not see how many times the cellular assay has been performed. In Figures 3 and 4. Average values have been calculated and standard deviation included indicating repetitions of the experiments. In the luminescence study is reported that the experiment was performed twice in duplicate. I would have preferred three times in triplicate.

All results included in the manuscript come from three independent experiments performed in triplicates. The description of the luminescent analysis methodology has been changed (from three times in duplicate (which we performed during screening analysis/method validation) to three times in triplicate (which we performed in experiments when the method is well validated). I apologize for the unnecessary confusion.

Saponin's main characteristic is that they hemolyze living mainly because they reduce surface tension. They can solubilize fats. This effect generally is nonspecific and may reduce the possibility of using saponin as a drug.  It is surprising that the authors have not mentioned this definition of saponins and have not called attention to the fact that the biological effect could be caused by damage to the cell membranes.

Thank you for this comment. We are of course aware that saponins are known for their haemolytic properties, which vary greatly depending on the structural features of the aglycone and sugar part. Recent data indicates that not only the functional groups and the glycidic moieties themselves, but their overall conformation affects haemolytic activity of saponins. The role of CIL1 fraction investigated in the current study can be compared to the use of saponins as adjuvants in vaccines. In the past it was claimed that that saponins should not be used as adjuvants in human vaccination due to their intrinsic haemolytic properties. Nevertheless, two of the highly toxic and haemolytic saponin fractions, obtained from Quillaja, namely Quil A and QS-21, have been extensively studied in pre-clinical experiments to finally develop a formulation that found its way to the clinics. QS-21 has recently been approved for use in human vaccines as a key component of combination adjuvants, e.g., AS01b in Shingrix® for herpes zoster. One of recent reviews on saponins as vaccine ad-juvants lists many efforts that were made to propose alternatives to QS-21 that can circumvent its drawbacks [Wang P.: Natural and synthetic saponins as vaccine adjuvants. Vaccines, 2021, 9, 222.]. According to your suggestion we have added a paragraph that discusses this issue in the Introduction. During the cytotoxic properties of CIL 1, we performed a cytotoxicity test based on the examination of cell membrane integrity (LDH test), the tested fraction of CIL1 did not cause damage to the cell membrane of cardiomyoblast cells, neuroblastomas, and liver cancer. Such a test may suggest that the fraction in the range of tested concentrations will not have permeabilizing activity. Moreover, investigations of the last two decades have shown that the observation – indeed true – that the amphiphilic saponins lyse cells must be clearly distinguished from highly specific effects due to functional groups. The latter one occurs at concentrations much lower than the first one and includes anti-inflammatory effects as hormone antagonists or anti-tumor effects as endosomal escape enhancer (Fuchs, H.; Niesler, N.; Trautner, A.; Sama, S.; Jerz, G.; Panjideh, H.; Weng, A. Glycosylated Triterpenoids as Endosomal Escape Enhancers in Targeted Tumor Therapies. Biomedicines 2017, 5, doi:10.3390/biomedicines5020014.).

It would be interesting to see if any single compound purified from the fraction CIL1 could mimic some of the effects described in this manuscript. 

We fully agree with Reviewer that purification of single compounds from fraction will be very interesting, and we will performed such analysis in the future. Other examples in the past had shown that the identification and complete purification of the active substance can take several years as shown for SA1641 as the main active substance in the mixture Saponinum album for Saponaria species. However, even in this case, the final efficacy of the active substance was not higher than in the mixture while some side effects in animal studies were reduced. The present work is the basis for the purification of a new endosomal escape enhancer. Future studies will compare different structures of active and non-active purified compounds to gain insight into structure-function relationships and derive a logical approach to chemical changes in the compound.

Reviewer 4 Report

The article entitled “Saponin fraction CIL1 from Lysimachia ciliata L. enhances the effect of a targeted toxin on cancer cells” describes the anti-cancer activity of fraction CIL1 against HER14 and NIH3T3 cell lines. The manuscript was written well.

However, this article does not meet the the requirements for novelty of Pharmaceutics journal.

Thus, I would not recommend this article for publication for the following reasons.

1. Section 1 -Introduction: should be improved considering the large number of works dealing with the subject of active biological compounds existing in Lysimachia ciliata L.

2. The saponin fraction CIL1 contains two compounds of interest, desglucoanagalloside B (Rt = 5.85 min) and a major compound eluting at Rt 5.12-5.14 min (29.2%). However, the manuscript do not provide the information of this peak.

Moreover, CIL1 comprises various minor compounds that collectively account for 30% of the fraction.  These compounds may influence the anti-cancer activity of CIL1, therefore the  identification of these compounds is required.

3. The anti-cancer activity of CIL1 may be affected by impurities in the fraction.

Thus, isolation of high purity compounds and investigation their anti-cancer activity are highly recommendation.

The current study is descriptive, with no indication as how the authors will develop further studies based on the results.

Minor editing of English language required. Some abbreviations are missing.

Author Response

The article entitled “Saponin fraction CIL1 from Lysimachia ciliata L. enhances the effect of a targeted toxin on cancer cells” describes the anti-cancer activity of fraction CIL1 against HER14 and NIH3T3 cell lines. The manuscript was written well.

However, this article does not meet the the requirements for novelty of Pharmaceutics journal.

Thus, I would not recommend this article for publication for the following reasons.

 We thank the reviewer for his opinion and regret that we could not convince him. As we had the chance to submit a revised version, we carefully addressed all of the critical points to further improve the manuscript. We are aware of the fact that the current work cannot address all the reviewer's concerns, but the aim of the present study was not to provide a finally purified substance for pharmaceutical use, but to introduce new promising ideas into the scientific discussion.

  1. Section 1 -Introduction: should be improved considering the large number of works dealing with the subject of active biological compounds existing in Lysimachia ciliata L.

Thank you for this comment. Yes, we agree that various compounds present in the plant species have biological activity. As the main scope of our current research was to evaluate how the addition of saponin fraction enhances the effect of targeted toxin, we felt that adding more information on other constituents would blur the main idea.

  1. The saponin fraction CIL1 contains two compounds of interest, desglucoanagalloside B (Rt = 5.85 min) and a major compound eluting at Rt 5.12-5.14 min (29.2%). However, the manuscript do not provide the information of this peak. Moreover, CIL1 comprises various minor compounds that collectively account for 30% of the fraction.  These compounds may influence the anti-cancer activity of CIL1, therefore the  identification of these compounds is required.

Thank you for this comment. In the characterization of saponin fraction denoted as CIL1 we have focused, at this stage of our experiments, on unequivocal identification of its main component. Mass fragmentation pattern of the major peak was identical to the compound previously isolated from this plant material and fully characterized structurally by 2D NMR as 3-O-β-D-xylopyranosyl-(1→2)-β-D-glucopyranosyl-(1→4)-[β-D-glucopyranosyl-(1→2)-]-α-L-arabinopyranosyl]-anagalligenin B, which is known under a trivial name desglucoanagalloside B. Other components of the mixture, most probably belong to the same class of 13β28-epoxy-oleananes, however at this stage, without their separation and isolation, only tentative structures could be suggested. In further stages of our research we aim to separate pure compounds, however, we did not go into detail here because the other peaks did not show any comparable effects in endosomal escape. Indeed, we do not know whether the main component of CIL-1 or another minor compound or a combination of them contributed to the observed effect. The main message of the current work is to demonstrate that the effect is present.

  1. The anti-cancer activity of CIL1 may be affected by impurities in the fraction.

Thus, isolation of high purity compounds and investigation their anti-cancer activity are highly recommendation. The current study is descriptive, with no indication as how the authors will develop further studies based on the results.

Other examples in the past had shown that the identification and complete purification of the active substance can take several years as shown for SA1641 as the main active substance in the mixture Saponinum album for Saponaria species. However, even in this case, the final efficacy of the active substance was not higher than in the mixture while some side effects in animal studies were reduced. The present work is the basis for the purification of a new endosomal escape enhancer. Future studies will compare different structures of active and non-active purified compounds to gain insight into structure-function relationships and derive a logical approach to chemical changes in the compound.

Reviewer 5 Report

The manuscript describes a study on enhancing the cytotoxicity of EGFR-targeted toxin – dianthin on HER14-targeted cells by a purified saponin fraction (CIL1) from Lysimachia ciliata L.

The research was adequately designed and executed. The manuscript was structured according to the Journal's requirements.

A few issues should be considered before the manuscript be ready for publishing.

1.       In the Introduction, the full structural name of the main saponin in CIL1 mixture as well as the reference to the source where its first isolation and identification from the title plant took place should be added.

2.       Does the hemolytic index CIL1 was established? Practically, the research findings are useless if the applied concentration of CIL1 could cause hemolysis. Please, elaborate on this in the text.

3.       Some tabulated data was detected in Figures 1 and 2. These should be removed from the figures and included in properly formatted tables. The reference should be given in the text.

4.       Some misformatting especially with superscripts and subscripts occurred in the text. The text should be carefully checked and corrected.

5.       References should be formatted and cited in the text according to the Journal style.

Author Response

The manuscript describes a study on enhancing the cytotoxicity of EGFR-targeted toxin – dianthin on HER14-targeted cells by a purified saponin fraction (CIL1) from Lysimachia ciliata L.

The research was adequately designed and executed. The manuscript was structured according to the Journal's requirements.

A few issues should be considered before the manuscript be ready for publishing.

  1. In the Introduction, the full structural name of the main saponin in CIL1 mixture as well as the reference to the source where its first isolation and identification from the title plant took place should be added.

Thank you for your valuable remark, as suggested, we added the structural name of CIL1 to introduction and reference where its chemical characterization and isolation took place.

  1. Does the hemolytic index CIL1 was established? Practically, the research findings are useless if the applied concentration of CIL1 could cause hemolysis. Please, elaborate on this in the text.

Thank you for this comment. We are of course aware that saponins are known for their haemolytic properties, which vary greatly depending on the structural features of the aglycone and sugar part. Recent data indicates that not only the functional groups and the glycidic moieties themselves, but their overall conformation affects haemolytic activity of saponins.

The role of CIL1 fraction investigated in the current study can be compared to the role of saponins as adjuvants in vaccines, their own activity is not as important as their enhancer effects. In the past it was claimed that saponins should not be used as adjuvants in human vaccination due to their intrinsic haemolytic properties. Nevertheless, two of the highly toxic and haemolytic saponin fractions, obtained from Quillaja, namely Quil A and QS-21, have been extensively studied in pre-clinical experiments to finally develop a formulation that found its way to the clinics. QS-21 has recently been approved for use in human vaccines as a key component of combination adjuvants, e.g., AS01b in Shingrix® for herpes zoster. One of recent reviews on saponins as vaccine adjuvants lists many efforts that were made to propose alternatives to QS-21 that can circumvent its drawbacks [Wang P.: Natural and synthetic saponins as vaccine adjuvants. Vaccines, 2021, 9, 222.]

As for CIL1 fraction used in our studies, we have at this stage not established its hemolytic index. According to you suggestion we have added additional paragraphs commenting this issue.

  1. Some tabulated data was detected in Figures 1 and 2. These should be removed from the figures and included in properly formatted tables. The reference should be given in the text.

As suggested by the Reviewer we removed table from figure 1 and 2, we included table in the text of manuscript and we give reference in the text.

  1. Some misformatting especially with superscripts and subscripts occurred in the text. The text should be carefully checked and corrected.

 We are very grateful for the Reviewer’s remark, the text of manuscript has been additionally checked and editorial errors have been removed.

  1. References should be formatted and cited in the text according to the Journal style.

Changed as recommended by the Reviewer.

Round 2

Reviewer 1 Report

The authors logically addressed the comments, provided additional information in the supplementary information, and promised to conduct recommended studies in the future. 

Reviewer 3 Report

The authors have satisfactorily answered my questions. 

Reviewer 4 Report

The authors have satisfactorily responded to comments raised by the reviewer. I am pleased to recommend the acceptance of this manuscript to Pharmaceutics.